# The Berlin Misophonia Questionnaire Revised (BMQ-R): Development and validation of a symptom-oriented diagnostical instrument for the measurement of misophonia

**Nico Remmert** *, **Katharina Maria Beate Schmidt, Patrick Mussel, Minne Luise Hagel, Michael Eid**

Department of Psychology, Freie Universität Berlin, Berlin, Germany

* n.remmert@fu-berlin.de

## Abstract

Misophonia is a clinical syndrome which is characterized by intense emotional and physical reactions to idiosyncratic sounds. However, its psychometric measurement is still in the early stages. This study describes the optimization of a self-report instrument, the Berlin Misophonia Questionnaire (BMQ), and addresses its strengths in comparison to existing psychometric measures. This new measure integrates contemporary empirical findings and is based on the latest criteria of misophonia. A cross-sectional online study was conducted using data of 952 affected as well as non-affected individuals. The final BMQ-R consists of 77 items in 21 scales, which were selected using a probabilistic item selection algorithm (Ant Colony Optimization). The results of confirmatory factor analyses, the assessment of reliability, and an extensive construct validation procedure supported the reliability and validity of the developed scales. One outstanding strength of the BMQ-R is its comprehensive measurement of misophonic emotional and physical responses. The instrument further allows for distinguishing between behavioral, cognitive, and emotional dysregulation; the measurement of clinical insight and significance; as well as discerning reactive and anticipating avoidance strategies. Our work offers several improvements to the measurement of misophonia by providing a reliable and valid multidimensional diagnostical instrument. In line with the scientific consensus on defining misophonia, the BMQ-R allows to formally recognize individuals with misophonia and so to compare findings of future studies. Undoubtedly, this measure fills a research gap, which we hope will facilitate the investigation of causes and treatment of misophonia.

## Introduction

Misophonia is a relatively new and still little investigated clinical syndrome that is characterized by severe affective, physiological, and behavioral symptoms triggered by the perception or anticipation of specific, typically human-induced sounds, for instance people eating or

**Data Availability Statement:** All data files are available from the Open Science Framework (doi. org/10.17605/OSF.IO/9VFMS).

**Funding:** The author(s) received no specific funding for this work.

**Competing interests:** The authors have declared that no competing interests exist.

drinking noisily [1–3]. Individuals with misophonia suffer from a decreased tolerance towards ordinary, innocuous sounds rather than sounds that are typically perceived as disturbing (e.g., microphone feedback [4]) [5]. Typical misophonic reactions to specific sounds include intense irritation, anger, distress, disgust, and anxiety [1, 6–8]. These reactions are relatively independent of acoustic characteristics (e.g., volume), they are mainly determined by the subjective meaning of sounds [2, 9] and by the contextual information involved [10]. Further, psychophysiological symptoms related to high levels of arousal and stress have been reported, for example, tachycardia, sweating, pressure, and impaired respiration [6, 11]. The intense response to specific sounds is accompanied by a perceived loss of control which potentially manifests itself as behavioral outbursts [1, 3]. The typical coping behavior is proactive avoidance of likely triggering situations or escaping such situations [1–3, 6]. The symptoms cause severe distress and functional impairment in individuals' interpersonal and occupational lives [1, 12, 13].

While leading researchers in the field of misophonia have reached a consensus on clinical characteristics [8], little is known about the differentiation from other disorders, its standing as a separate disorder, causes, and treatment [2, 6]. The major reason for this research gap is the lack of psychometrically robust and validated measurement instruments. To date, there is no measure that allows to diagnose misophonia and to identify individuals in need of clinical treatment. We hence argue that there is a scientific as well as practical need for a diagnostical instrument that allows to formally recognize individuals with misophonia according to diagnostic criteria.

To attain a mutual understanding of misophonia, Schröder et al. [3] proposed the first set of diagnostic criteria for misophonia, which were revised and validated in a follow-up study ([1], see Table 1 for the revised criteria). Despite the raised criticisms [e.g., 14, 15] which imply further refinement of the criteria, we believe that the proposed criteria are a good starting point for the development of empirically verifiable measurement models and instruments. It is further evident that the criteria largely reflect the consensus definition of misophonia [8] and, most importantly, they formalize the understanding of misophonia which allows future studies to be compared with each other.

**Table 1. Revised diagnostic criteria for misophonia.**

| Criterion description |
| --- |
| A. Preoccupation with a specific auditory, visual or sensory cue, which is predominantly induced by another person. It is required that oral or nasal sounds are a trigger. |
| B. Cues evoke intense feelings of irritation, anger and/or disgust of which the individual recognizes it is excessive, unreasonable or out of proportion to the circumstances. |
| C. Since emotions trigger an impulsive aversive physical reaction, the individual experiences a profound sense of loss of self-control with rare but potentially aggressive outbursts. |
| D. The individual actively avoids situations in which triggers occur or endures triggers with intense discomfort, irritation, anger or disgust. |
| E. The irritation, anger, disgust or avoidance causes significant distress and/or significant interference in the individual's day-to-day life. For example, it is impossible to eat together, work in an open office space or live together. |
| F. The irritation, anger, disgust and avoidance are not better explained by another disorder, such as an Autism Spectrum Condition (e.g. a general hypersensitivity or hyper arousal to all sensory stimuli) or Attention Deficit Hyperactivity Disorder (e.g. attention problems with high distractibility in general). |

The listed criteria are adopted according to Jager et al. [1].

## Existing instruments for measuring misophonia

Although some frequently used instruments exist and several new instruments for measuring misophonia have only recently been developed, major psychometric limitations and shortcomings remain.

The most frequently used instruments are the Amsterdam Misophonia Scale (A-MISO-S; [3]) and the Misophonia Questionnaire (MQ; [12]). The A-MISO-S is an instrument that was constructed in line with the proposed diagnostic criteria. However, the authors did not report any psychometric properties of the instrument, neither in its original nor in its revised version (AMISOS-R; [1]). The Misophonia Questionnaire is a self-report instrument developed to measure reactions to sounds as well as the severity of sound sensitivity. Besides pending validation studies in samples with afflicted individuals, the measure is criticized for being too broad and is thus not necessarily a specific measure of misophonia [16].

Among the most recently published instruments are the MisoQuest [16], the Selective Sound Sensitivity Syndrome Scale (S-Five; [15]), and the Duke Misophonia Questionnaire (DMQ; [17]. The MisoQuest is a unidimensional questionnaire based on the criteria proposed by Schröder et al. [3]. Good psychometric properties have been shown for the instrument [16]. However, not all relevant misophonic reactions are considered, as the MisoQuest for instance solely measures anger reactions. Also, the diagnostic criteria proposed by Schröder et al. [3] are unlikely to be unidimensional as they comprise different facets of misophonia [18]. Due to its small number of items and its unidimensional structure, the MisoQuest can thus be considered a screening instrument.

The multidimensional Selective Sound Sensitivity Syndrome Scale (S-Five; [15]). measures different cognitive, emotional and behavioral symptoms of misophonia in five subscales. Extensive examinations demonstrated the instrument to have good psychometric properties. However, the S-Five is limited in its scope as it only models the perceived threat when individuals cannot avoid specific sounds, neglecting different avoidance strategies, which are important characteristics of misophonia [1, 2, 6]. Moreover, the S-Five does not distinguish between different aspects of dysregulation and omits dimensions of clinical insight.

So far, the DMQ is the most comprehensive scale with nine dimensions covering a wide range of misophonic symptoms and the authors provided a profound validation study demonstrating good psychometric properties. Still, some shortcomings are noteworthy: Affective responses are not differentiated, coping is measured in broad categories referring to the time misophonic triggers occur (instead of differentiating specific coping strategies such as avoidance), and dimensions of clinical insight are omitted. So although some initially validated instruments have already been developed, so far, there is no diagnostic instrument measuring the entirety of relevant symptoms and criteria. The literature evidently shows that there is a lack of a theory-based diagnostic instrument. Recognizing the importance of a psychometrically sound and comprehensive measurement of misphonia on the one hand and the lack of such such an instrument on the other hand, the Berlin Misophonia Questionnaire (BMQ) was developed.

The Berlin Misophonia Questionnaire (BMQ; [18]) is a German self-report instrument to measure misophonic symptoms originally based on the diagnostic criteria by Schröder et al. [3]. It comprises two parts: (a) a condition part, measuring the extent of disturbance attributed to internally or externally produced sound classes (i.e., groups of sounds which share subjective meaning) and general sound intolerance symptoms (henceforth denoted as 'GSIS'), and (b) the symptoms part with six scales which correspond to the diagnostic criteria. The deductive construction rationale targeted the development of items for each of the six defined criteria in a reflective latent measurement model. A psychometric analysis revealed good to

excellent reliabilities of the subscales and mostly acceptable measurement models [18]. The instrument's convergent and discriminant validity was supported by variance-analytical and correlative results [18]. However, the measurement models were found to still require several optimizations, and approaches for further development were suggested.

## The present study

The present study aimed to adress the outlined research gaps regarding the assessment of misophonia by further developing and optimizing the BMQ. For this purpose, a revised version of the BMQ, the Berlin Misophonia Questionnaire Revised (BMQ-R), was created. The BMQ-R incorporates contemporary empirical findings and new approaches to modelling criteria of misophonia. The revised diagnostic criteria by Jager et al. [1] provided the basis for this advancement. A symptom-oriented approach allowed to delineate sub-aspects of the criteria and hence realize a clear measurement of all symptoms. Further psychometric properties of the BMQ-R scales were investigated in an extensive validation study. Finally, the study aimed to provide a better comprehension of the multidimensional nature of misophonic symptoms in form of a nomological network and to give insight into the validity of the diagnostic criteria.

In the following, the symptom-oriented modelling approach applied in the development of the BMQ-R is described in detail. The symptom-oriented scales resulting from the modeling approach enable the derivation of several hypotheses that are used to test the instrument's construct validity. The construct validation procedure including these hypotheses is, too, described below.

**Symptom-oriented modeling approach.** In their criteria, Jager et al. [1] describe an aversive emotional and physical reaction to the presence or anticipation of specific sounds. Therefore, in the construction of the BMQ, two separate models were specified for the reaction to present and anticipated sounds, respectively. Additionally, modification indices of the first version of the BMQ [18] as well as theoretical considerations suggested a distinction between specific emotional and physical responses for both subscales [1, 6, 7, 9]. Items representing either anger or physical reactions demonstrated substantial residual correlations. This distinction has also been shown in an *a posteriori* explorative factor analysis of the data from the initial development study [18]. A symptom-oriented approach was hypothesized to better delineate sub-aspects of the aversive emotional and physical response.

Jager et al. [1] list irritation, anger, disgust, and physical reactions as different aspects of the misophonic response. We thus specified each specific response separately in different measurement models so that subscales for irritation, anger, disgust, anxiety, and physical reactions were differentiated, each subdivided into anticipatory and present. Beside a clearer measurement of the criteria, the benefit of this approach is the option to regard accessory symptoms.

Jager et al. [1] further describe a *general* loss of self-control with potential aggressive outbursts in their criteria. A recent study, however, has shown that different facets of dysregulation are correlated with overall misophonic symptoms [19]. This suggested an extension of the previously proposed general loss of control. We therefore distinguish cognitive, emotional, and behavioral dysregulation in the measurement of misophonia. In this vein, aggressive outbursts are predominantly incorporated in the facet behavioral dysregulation.

To further consider the accessory nature of symptoms defined by Jager et al. [1], formerly jointly measured symptoms are measured separately. This includes the criteria B. (recognition of excessive, unreasonable or disproportionate reactions to cues) and E. (significant distress or interference in the daily life) (for details see Table 1) being separated into (1) recognition of excess and (2) recognition of disproportionality as well as (3) significant distress and 4) interference in the individual's daily life (functional impairment).

A symptom-oriented optimization is implemented regarding avoidance behavior so that avoidance is separated into (1) reactive avoidance and (2) anticipatory avoidance. Reactive avoidance involves quitting the situation and the reduction or prevention of auditory perception when being confronted with triggers. In contrast, anticipatory avoidance is defined by safety behavior like preparatory avoidance of potential sounds. Although this distinction is not made in the criteria by Jager et al. [1], when describing symptoms of mental disorders, there is a common distinction between escape and avoidance [20].

The symptom-oriented approach increased the number of subscales, but in return allows for a much clearer assessment of misophonia considering each symptom. Moreover, the approach is an improvement to the conventional usage of sum scores or G-factors in clinical measures (e.g., [21]). An outline of the proposed diagnostic model and symptom-oriented scales of the BMQ-R can be found in Fig 1.

**Construct validation procedure.** To continue and extend the former BMQ's construct validation, based on the nomological network of the constructs, we stated specific theory-driven or literature-based validation hypotheses, all of which are presented below.

**Aversive reactions to the presence or anticipation of sounds.** A central aim of the validation was to differentiate aversive misophonic reactions. Despite being partially similar [22], anger and irritation were modeled as distinct but correlated constructs since current literature suggests their differentiation [23]. In contrast to anger, Toohey and DiGiuseppe [24] define irritation as a reactive mood characterized by physiological arousal and increased sensitivity to sensory stimuli, especially typically less disturbing stimuli. We thus hypothesize misophonic anger reactions to be strongly correlated with misophonic irritation reactions. Moreover, a

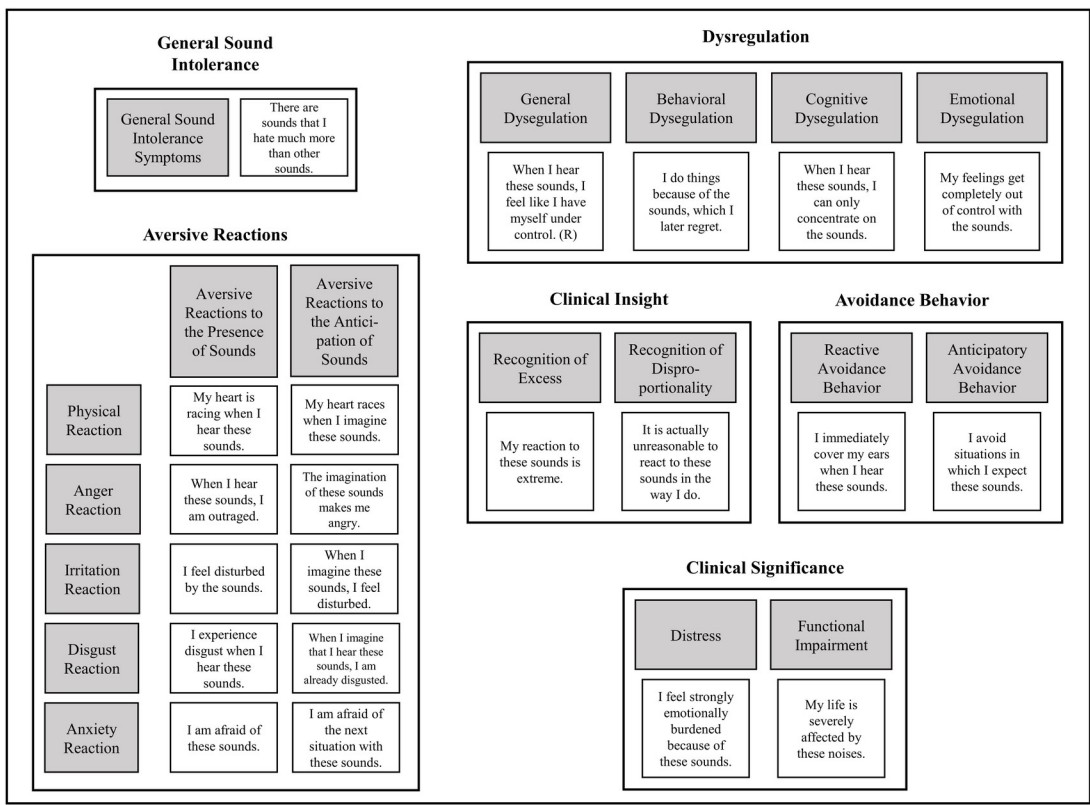

**Fig 1. Outline of the symptom-oriented scales and exemplary items.**

strong positive correlation of misophonic irritation with misophonic physical reactions is expected. However, we assume convergent irritation measures to display a larger positive correlation with misophonic irritation than with misophonic anger. Convergent anger measures, in turn, were assumed to correlate more strongly positively with misophonic anger than with misophonic irritation. Further, we hypothesized moderate to strong positive correlations between misophonic anger, anger dispositions, aggressive outbursts, external appraisals, verbal, and physical aggression as well as hostility. Hostility has also been demonstrated to strongly correlate with irritation [23], so we likewise assumed this relationship for misophonic irritation reactions.

Regarding misophonic disgust, we expected strong positive correlations with the respective convergent measure of disgust. For misophonic anxiety, we expected strong positive correlations with both somatic and cognitive anxiety symptoms.

Since the physical misophonic reaction is not specific to any emotional response, strong positive correlations with each emotional misophonic response were hypothesized. Additionally, somatic anxiety symptoms were assumed to show a strong positive correlation with physical misophonic responses.

**Clinical insight.**   Recognition of excess and disproportionality is referred to the detection and labelling of mental states and behavior as pathological [25], here pertaining to misophonia. This conceptual classification of insight is defined to be a meta-cognitive process [26] and an aspect of self-knowledge involving information about defense mechanisms and coping strategies [27]. Therefore, we hypothesized recognition of excess and disproportionality to be positively correlated with illness coherence. Illness coherence is defined as the extent to which an individual comprehends their illness [28]. Individuals who tend to recognize excessive or disproportional behavior were hence assumed to better comprehend their misophonia symptoms. Further, individuals' internal and external appraisals of misophonic symptoms (i.e., their symptom attribution) were assumed to display a strong positive correlation with the recognition of excess and disproportionality, whereas external appraisals were expected to be less correlated since misophonia is considered ego dystonic [29].

**Dysregulation.**   General misophonic dysregulation and its specific subfacets (behavioral, cognitive, and emotional dysregulation) were assumed to positively correlate with convergent general dysregulation and difficulties in impulse control, difficulties in goal-directed behavior, and limited access to emotion regulation strategies. Since limited access to emotion regulation strategies, in terms of content, reflects emotional dysregulation, it was hypothesized to correlate more strongly with misophonic emotional dysregulation than with other subfacets. Further, difficulties in impulse control were assumed to correlate more strongly with behavioral misophonic dysregulation and misophonic outbursts, whereas difficulties in goal-directed behavior were assumed to correlate more strongly with cognitive misophonic dysregulation.

**Avoidance behavior.**   We hypothesized reactive and anticipatory misophonic avoidance behavior to be strongly correlated with a convergent measure of avoidance of sounds in daily life. Further, behavioral experiential avoidance was assumed to have a strong positive correlation with reactive and anticipatory avoidance behavior. Additionally, we assume that the more individuals experience intense aversive emotional and physical responses to the presence of sounds, the more they show misophonic avoidance behavior since this is a central coping mechanism [1, 6]. This was especially assumed for reactive avoidance in contrast to anticipatory avoidance which was hypothesized to be reflected through lower correlations of aversive emotional and physical reactions with anticipatory avoidance behavior.

**Clinical significance.**   Misophonic distress was assumed to strongly correlate with aversive emotional and physical misophonic responses as well as with convergent misophonic impact and overall symptom burdening. Misophonic functional impairment was hypothesized to

strongly correlate with different dimensions of functional impairment defined by the World Health Organization [30] (cognition, interaction with others, life activities, household, and participation in social life) and overall symptom burden.

## Materials and methods

### Study overview

The study was divided into two parts. First, we translated and pretested some of the validation instruments since neither German versions nor equivalent alternatives were available. At the same time, we developed and pretested revised sound classes for the measurement of sound disturbance. Secondly, the BMQ-R was jointly administered with the translated and additional validation instruments in the main study which we describe in the following. Further information on the pretest is available in S1 Text.

Data protection guidelines were met, and participants gave informed consent before completing the survey. The study was approved by the Ethics Committee at the Department of Education and Psychology of the Freie Universität Berlin, Germany (document number: 029/2020).

### Development of the item pool

For the symptoms part, we tested 120 items in total. These items are partly taken from the first version of the BMQ, which consists of 42 items [18]. An additional set of 21 items from the former item pool were selected to ensure an adequate length of the symptom-oriented scales. Moreover, 57 new items were constructed since the original item pool did not include enough items for each scale.

The original items of the GSIS scale were rephrased to slightly increase item difficulty as indicated in former analyses [18]. Also, the items from the symptoms part were rephrased to generically refer to sounds in plural instead of referring to a specific sound.

The first version of the BMQ contained two additional scales measuring the extent to which individuals feel burdened by and sensitive to different sound classes being *externally* or *internally* produced (e.g., "people eating" or "nasal sounds"). Qualitative results from the first study on the BMQ [18] suggested a revision of the sound classes preventing an overlap of the different classes. The revised sound classes were included in the pretest of this study (see S1 Text).

### Measures

**Berlin Misophonia Questionnaire Revised (BMQ-R).** Like the BMQ [18], the BMQ-R is divided into two parts. The condition part consists of 21 items measuring the disturbance of internally or externally produced sound classes and four items measuring GSIS which are rated on a 6-point rating scale (0 = *does not apply at all* to 5 = *completely applies*). The symptom part comprises 20 symptom-oriented scales. In total, the symptom part comprises 120 items which are rated on a 6-point rating scale (0 = *does not apply at all* to 5 = *completely applies*), whereas items on the disturbance of sound classes are rated on a different 6-point rating scale (0 = *not disturbing at all* to 5 = *maximally disturbing*).

**Selective Sound Sensitivity Syndrome Scale (S-Five; [15]): Experiences Section (S-Five-E).** The S-Five comprises two sections: the experiences (S-Five-E) and the feelings (S-Five-F) section [31]. Only the S-Five-E section was included, which consists of 25 items in five subscales: internalising appraisals, externalising appraisals, sense of emotional threat, outbursts, and impact. The items are rated on an 11-point rating scale (0 = *not at all true* to 10 = *completely true*). The S-Five-E was translated to German using the TRAPD procedure [32]. For more information on this procedure, see the description and results of the pretest. Good

internal consistency estimates have been reported for the subscales (α = .83 - .88) and the measure demonstrates diverse evidence on construct validity, for example a strong correlation with a convergent measure of misophonia (*r* = .64) [15].

**MisoQuest [16].** Based on the diagnostic criteria by Schröder et al. [3], the MisoQuest is a screening instrument of misophonia with 14 items. Agreement to each item is rated on a 5-point rating scale (1 = *strongly disagree* to 5 = *strongly agree*).The scale was translated to German using the TRAPD procedure [32]. The estimated internal consistency of the scale is excellent (α = .96) [16]. Further, differential validity is substantiated by large mean differences on the MisoQuest between individuals with and without misophonia (*d* = 2.13) [16].

**Aggression Questionnaire (AQ; [33]).** The German version of the original Aggression Questionnaire was used [34]. Four dimensions of aggression are operationalized: 1) physical and 2) verbal aggression, 3) anger and 4) hostility. The scale comprises 29 items which are rated on a 4-point rating scale (1 = *does not apply* to 4 = *fully applies*). from the range of internal consistency estimates is .62 ≤ α ≤ .82 [34]. Regarding evidence on construct validity, the scales correlated highly with a convergent measure of anger expression (*r* = .56 - .73) [34].

**Disgust Propensity and Sensitivity Scale Revised (DPSS-R; [35]): Disgust propensity scale.** The DPSS-R reduced-item version [35] is a measure of disgust with the dimensions disgust propensity (henceforth denoted as 'DP') and disgust sensitivity. It consists of 12 items regarding the frequency of physical and emotional symptoms of disgust which are rated on a 5-point rating scale (1 = *never* to 5 = *always*). For this study, only the DP items were used (six items). DP reflects how easily an individual is disgusted The scale was translated to German using the TRAPD procedure [32]. The internal consistency of the DP scale is good (α = .83) [35]. Correlative results demonstrate evidence for concurrent validity regarding disgust-relevant symptoms of phobias (*r* = .32 - .39) [35].

**State-Trait Inventory for Cognitive and Somatic Anxiety (STICSA; [36]).** The STICSA is a measure of cognitive and somatic state trait anxiety. Only the 21 trait items were used which are rated on a 4-point rating scale (1 = *not at all* to 4 = *very much so*). The inventory differentiates two factors: The first factor reflects cognitive symptoms (10 items), whereas the second factor reflects somatic symptoms of anxiety (11 items). Both factors have good internal consistency (α = .88) [36]. The scales was translated to German using the TRAPD procedure [32]. There is supporting evidence for the validity of the inventory, for example high correlations with a similar state anxiety measure (*r* = .53 - .63) [37].

**Brief Irritability Test (BITe; [23]).** The BITe is a 5-item measure of irritability minimalizing the innate overlap with related constructs (e.g., anger or depression). Items are rated on a 6-point rating scale (1 = *never* to 6 = *always*). For this study, the German version was used [38]. The BITe has good internal consistency (α = .88) and evidence on convergent validity (e.g., high correlations with self-report scales of irritability) [23].

**Difficulties in Emotion Regulation Scale (DERS; [39]).** The DERS is a multidimensional measure of emotion regulation comprising six subscales, three of which were used in the present study: 1) impulse control difficulties, 2) difficulties engaging in goal-oriented behavior, and 3) limited access to emotion regulation. These subscales consist of 19 items in total, 15 of which were chosen regarding their content validity to match the intended purpose. The items are rated on a 5-point rating scale regarding the experienced frequency (1 = *almost never* (0–10%) to 5 = *almost always* (91–100%)). We used the German version [40], the scales of which possess good internal consistency (α = .88 - .91). Further, evidence for construct validity has been shown through medium to high correlations with internalising or externalising problem behavior in their respective direction (*r* = .29 - .64) [40].

**Noise Avoidance Questionnaire (NAQ; [41]).** The NAQ is a German self-report instrument measuring the avoidance of sounds in daily life. It comprises 25 items, 10 of which state

specific situations that might be avoided (e.g., restaurants or concerts). The remaining 15 items refer to specific behaviors related to sound avoidance. Only the more behavior-oriented items were chosen, which are rated on a 5-point rating scale (0 = *never* to 6 = *very often/ always*). The English items were naïvely translated. Since the statements are short and concise, no compromising translational effects were assumed. The scale possesses good psychometric properties. Internal consistency is excellent ($\alpha$ = .96) and evidence on convergent validity has been demonstrated through high correlations with a measure of sound intolerance ($r$ = .61) [41].

**Brief Experiential Avoidance Questionnaire (BEAQ; [42]).** The BEAQ is a 15-item uni-dimensional measure of experiential avoidance behavior and a reduced-item version of the Multidimensional Experiential Avoidance Questionnaire (MEAQ; [43]). For this study, only items from the former Behavioral Avoidance subscale of the German version [44] were chosen which reflect situational avoidance of physical discomfort and distress. The respective items are rated on a 6-point rating scale (1 = *strongly disagree* to 6 = *strongly agree*). The internal consistency of the BEAQ is good ($\alpha$ = .87) and it correlates highly with convergent measures of avoidance [44].

**World Health Organization Disability Assessment Schedule 2.0 [30].** The WHODAS 2.0 is a generic clinical instrument based on the International Classification of Functioning, Disability and Health (ICF; [45]) which measures the impact of a given health condition in terms of functioning in six domains of life: Cognition, mobility, self-care, getting along, life activities, and participation. Mobility and self-care appear to be irrelevant for misophonia. The German self-report version was used, containing 27 out of the initial 36 items after discarding the two domains. The degree of difficulty individuals have performing the indicated activities is rated on a 5-point rating scale (1 = *none* to 6 = *extreme or cannot do*). Internal consistency of the relevant domains in mental health applications is excellent ($\alpha$ = .92 - .94) [30]. There is evidence for convergent validity in the form of high correlations with construct-related measures [30].

**Illness Perception Questionnaire Mental Health (IPQ-MH; [46]): Illness coherence scale.** The IPQ-MH is an adapted version of the Illness Perception Questionnaire Revised (IPQ-R; [28]) measuring individual's perceptions of their mental health problem. The Illness Coherence scale used in the study comprises five items measuring the extent of an individual's comprehension regarding their mental health problem. Items are rated on a 5-point rating scale (1 = *strongly disagree* to 5 = *strongly agree*). The German version of the IPQ-R [47] was used and adapted in line with Witteman et al. [46] by replacing the term '*illness*' with '*problem*' in each item. Good internal consistency is reported ($\alpha$ = .83) [45]. Further, illness coherence is positively correlated with stigma indifference in patients with obsessive-compulsive disorder ($\tau$ = .24) [48].

## Procedure

The study comprised two tracks to which participants were assigned according to a specified cutoff of the GSIS scale from the BMQ-R. Participants with a sum score greater than 12 completed the track for potentially affected individuals. The cutoff was chosen pragmatically, considering that a sum score of 12 means average agreement on the four items. In this track, participants were instructed to think about the most disturbing or burdensome noises when filling out the survey. We assumed that individuals who mainly disagree with general sound intolerance items (GSIS sum score < 13) are likely to not have a cognitive representation of disturbing or burdening sounds or are incapable of designating such sounds. Thus, we instructed those participants to think about people eating noisily, swallowing, and sniffing,

which were the sounds people affected by misophonia most frequently mentioned in a previous study [18]. The instructions also restricted participants to only consider auditive information of the sounds, since so far, the impact other sensory information have on misophonia symptoms has not been substantiated sufficiently. Items were presented randomly in their respective content area. To ensure that a) participants consistently consider the presence or impact of the chosen or given sounds and that b) generic validation instruments match the misophonic contextualization, we instructed participants to consider those sounds in relation to the respective scale (e.g. by adding a frame of reference like "[. . .] when you are confronted with bothersome sounds"). This contextualization minimizes within-person inconsistency and between-person variability through using the frame of reference, leading to higher reliability and validity when responding to generic items [49].

## Inclusion criteria and item omissions

We included affected as well as non-affected individuals aged 16 or older. To ensure data quality, we presented items to assess the participants' attention and only included individuals who correctly answered at least 80% of them [50]. Further, we checked for consecutive response patterns (response pattern index; [51]), excluding participants with more than 30% consecutive answers.

Item omissions were assumed to be missing at random (MAR). Hence, we addressed omissions by using full information maximum likelihood estimation in specified measurement models whenever applicable. However, for pragmatic reasons, we decided to exclude participants with more than 50% missing responses for at least one scale. Thus, model specification consistently incorporated the maximal number of participants available according to the exclusion criteria (i.e., the number of participants varied by scale as indicated).

## Model specification, model fit evaluation, and reliability estimation

First, the subscales of the BMQ-R were jointly modeled according to the respective content area (in line with the diagnostic criteria). Hence, correlated first-order factor models were specified for the respective symptom-oriented scales. Specified models were optimized with respect to the total number of items and model fit with different item selection algorithms (see item selection procedure). Subsequently, a complete model with all symptom-oriented scales was specified to evaluate model fit and to check for cross-loadings, thus ensuring a clear distinction of the scales.

Secondly, validation instruments were modeled according to the results of the pretest or original factor structure. For unidimensional scales, item selection procedures were applied to achieve acceptable model fit. By doing so, we carefully considered minimalizing the loss of content through item reduction. All deviations from the original scales are described in the results.

To take non-normality and categorical indicators into account, the weighted least square mean and variance adjusted (WLSMV) estimator with ordered categories was used. For continuous indicators we used maximum likelihood estimation with robust (Huber-White) standard errors [52, 53] and a scaled test statistic to address nonnormality of the data. The R package "lavaan" (version 0.6–9; [54]) was used for all model specifications.

Model fit was evaluated using $\chi^2$-tests and common cutoff values for fit indices (RMSEA close to .06, SRMR close to .08, Mc (McDonald's Centrality) close to .90 [55]) as well as a CFI (Comparative Fit Index) close to .97 and a TLI (Tucker-Lewis Index) close to .97 [56]. For models with ordinal indicators, the scaled model fit statistics were used. Moreover, ECVI (Expected Cross Validation Index) was inspected to compare non-nested models [56].

The scales' reliabilities were estimated using model-based McDonald's ω [57]. Further, 95% confidence intervals were calculated via bias-corrected and accelerated bootstrap with a bootstrapping sample size of $B = 1000$.

## Item selection procedure

Items were selected using a probabilistic item selection algorithm (Ant Colony Optimization (ACO) algorithm; [58]). This algorithm is based on the food foraging behavior of ants harnessing virtual "pheromones" to guide the selection towards solutions with good psychometric properties. An adaption of the *MAX-MIN* Ant System by Stützle and Hosos [59] was used as implemented in the R package "stuart" (version 0.9.1–9000, [60]). Moreover, a hold-out-validation ACO algorithm [61] with a modified model fit criteria function was applied. Data was split in half into a calibration and a validation sample. With respect to the model fit criteria function, we used a combination of the RMSEA, SRMR, and CFI unless otherwise specified. The benefit of this ACO-approach is that the developed models are robust and economic. When possible, a brute force algorithm was used which allows to evaluate every possible item combination to obtain the best combinatorial solution.

## Participants

We based our sample sizes on Monte Carlo simulations performed in previous studies [62]. Yuan and Bentler [62] suggest a sample size greater than 400 for continuous nonnormal data with missing responses and MLR estimation. A similar reference point was suggested by Forero et al. [63] for ordinal data and WLSMV estimation ($N \geq 200$–500). Participants were recruited in Germany via online posts on social media (Facebook and Instagram) and university mailing lists. Specific groups of individuals identifying as having misophonia as well as unspecific groups were chosen. Psychology students received course credit as an incentive. After excluding participants according to the defined criteria, the total sample comprised $N = 952$ individuals who completed the first GSIS scale of the BMQ-R. $N = 621$ (65.2%) individuals completed the symptom part of the BMQ-R and $N = 601$ completed the whole survey (63.1%).

Most participants (86.7%) were female, and seven individuals indicated non-binary gender. The mean age was $M = 33.85$ years ($SD = 11.19$, range 16 to 69). 29.9% of all participants were students. Further, 32.7% had a university degree and 45.5% had at least a high school equivalent qualification. Most participants had a partner or were married (66.6%), whereas 27.0% did not have a partner or were living separately. 30.1% were employed either part-time or marginally, 38.7% were employed full-time, and 12.0% were unemployed.

## Results

### Factorial validity

**General sound intolerance and sound classes.** Only 6.7% of the participants had a GSIS sum score lower than 13 (chosen score for non-affected individuals), whereas more than half of the sample indicated the highest score of 20 (49.8%). The mean GSIS sum score was $M = 18.05$ ($SD = 3.13$) with males indicating lower scores ($t(131.64) = -2.99$, $p < .01$). These results suggest that the sample mainly comprised potentially affected individuals. The three most disturbing sound classes were "people eating" ($M = 3.89$, $SD = 1.60$), "nasal sounds" ($M = 3.34$, $SD = 1.64$), and "drinking people" ($M = 3.20$, $SD = 1.84$), whereas "animal sounds" were considered the least disturbing sound class ($M = 0.98$, $SD = 1.46$).

**Table 2. Results of confirmatory factor analyses for the symptom part of the BMQ-R.**

| Model | $\chi^2_s$ | df | $\frac{\chi^2_s}{df}$ | $CFI_s$ | $TLI_s$ | $RMSEA_s$ (90%-CI) | SRMR | Mc | ECVI |
|---|---|---|---|---|---|---|---|---|---|
| | | | | | | Goodness of fit statistics | | | |
| Aversive Reactions to the Presence of Sounds–Five-factor correlated model | 637.20*** | 142 | 4.49 | .98 | 0.97 | .07 (.06 - .07) | .05 | 0.85 | 0.83 |
| Aversive Reactions to the Anticipation of Sounds–Five-factor correlated model | 68.54*** | 25 | 2.74 | 1.00 | 1.00 | .05 (.04 - .07) | .02 | 1.00 | 0.26 |
| Clinical Insight–Two-factor correlated model | 68.84*** | 19 | 3.62 | 1.00 | 1.00 | .06 (.05 - .08) | .02 | 1.00 | 0.18 |
| Dysregulation–Four-factor correlated model | 384.29*** | 98 | 3.92 | .99 | 0.98 | .06 (.06 - .07) | .03 | 0.93 | 0.57 |
| Dysregulation–Bifactor S-1 model | 298.65*** | 89 | 3.36 | .99 | 0.99 | .06 (.05 - .07) | .03 | 0.97 | 0.51 |
| Avoidance Behavior- Two-factor correlated model | 54.66*** | 19 | 2.88 | 1.00 | 1.00 | .05 (.04 - .07) | .03 | 1.00 | 0.18 |
| Clinical Significance–Two-factor correlated model | 91.18** | 53 | 1.72 | 1.00 | 1.00 | .03 (.02 - .05) | .02 | 1.01 | 0.31 |
| Joint Model–Fifteen-factor correlated model[a] | 3303.80*** | 1785 | 1.85 | .98 | 0.98 | .04 (.04 - .04) | .04 | 0.59 | 5.87 |

$N$ = 611–789. $\chi^2_s$ = scaled $\chi^2$-value; $CFI_s$ = scaled Comparative Fit Index; $TLI_s$ = scaled Tucker-Lewis Index; $RMSEA_s$ = scaled Root Mean Square Error of Approximation; SRMR = Standardized Root Mean Square Residual; ECVI = Expected Cross Validation Index; Mc = McDonald's Centrality Index.

[a] $N$ = 575.

** $p < .01$.

*** $p < .001$

The GSIS scale showed good fit to the data ($\chi^2_s(2)$ = 17.91, $p < .01$, $CFI_s$ = 1.00, $TLI_s$ = .99, $RMSEA_s$ = .07 [.04 - .11], SRMR = .02, Mc = 1.00). The reliability was estimated at $\omega$ = .84.

**Aversive reactions to the presence of sounds.** The hold-out-validation ACO algorithm yielded 19 items (out of 28) in five subscales. The selected model showed good fit to the data and was strongly invariant between calibration and validation sample ($\Delta\chi^2(95)$ = 99.06, $p$ = .37). Results of the factor analyses for the symptom part are given in Table 2. The subscales had good to excellent reliabilities at $.85 \leq \omega \leq .91$, except for irritation reaction ($\omega$ = .77). Reliability estimates for all scales of the BMQ-R are given in Table 3.

**Table 3. Reliability estimates for the BMQ-R scales.**

| Model/Scale | ω (95%-CI) | Model/Scale | ω (95%-CI) |
|---|---|---|---|
| General Sound Intolerance Symptoms (GSIS)[a] | .84 (.81- .87) | Recognition of Disproportionality | .90 (.88 - .92) |
| Aversive Reactions to the Presence of Sounds | | Dysregulation | |
| Anger Reactions | .90 (.89 - .91) | General Dyregulation | .91 (.89 - .92) |
| Irritation Reactions | .77 (.73 - .81) | Behavioral Dysregulation | .86 (.83 - .88) |
| Disgust Reactions | .91 (.90 - .92) | Cognitive Dysregulation | .90 (.87 - .91) |
| Anxiety Reactions | .88 (.86 - .90) | Emotional Dysregulation | .87 (.85 - .88) |
| Physical Reactions | .85 (.82 - .87) | Avoidance Behavior | |
| Aversive Reactions to the Anticipation of Sounds | | Reactive Avoidance | .72 (.67- .75) |
| Anger Reactions | .93 (.91 - .94) | Anticipatory Avoidance | .93 (.92- .94) |
| Irritation Reactions | .87 (.84 - .89) | Clinical Significance | |
| Disgust Reactions | .87 (.84 - .89) | Distress | .94 (.93- .95) |
| Anxiety Reactions | .90 (.87 - .92) | Functional Impairment | .92 (.90- .93) |
| Physical Reactions | .84 (.80 - .87) | | |
| Clinical Insight | | | |
| Recognition of Excess | .91 (.90 - .92) | | |

$N$ = 611–789. $\omega$ = McDonald's Omega. 95%-confidence intervals were calculated via bias-corrected and accelerated bootstrapping with a bootstrapping sample size $B$ = 1000.

**Aversive reactions to the anticipation of sounds.** Since the anticipation of sounds elicits less aversive reactions than the presence of sounds [18] and therefore tends to be rather subordinate in the assessment of misophonic reactions, we decided to develop an ultra-short model comprising 10 items in five subscales through a hold-out validation brute force algorithm. The model demonstrated excellent fit and was strongly invariant between calibration and validation sample, but some intercorrelations were very high, which indicated overlapping constructs. Hence, a second ACO algorithm with an adapted criterion function minimizing the relevant intercorrelations was applied. The selected model showed good fit to the data and was still strongly invariant across the hold-out validation samples ($\Delta\chi^2(50) = 45.05$, $p = .67$). Reliability estimates were good to excellent for all subscales at $.84 \leq \omega \leq .93$.

**Clinical insight.** A hold-out validation brute force algorithm was applied to select four items in two subscales, respectively (8 out of 12 in total). The solution demonstrated excellent fit to the data but was only weakly invariant across the hold-out validation samples ($\Delta\chi^2(6) = 3.22$, $p = .78$). Reliability estimates for the two scales were $\omega = .90$ and $\omega = .91$.

**Dysregulation.** Four items per subfacet and four items for the general dysregulation facet (16 of 27 items in total) were selected using the ACO algorithm. The selected items were then modeled in a correlated first-order factor model. This model demonstrated good fit and was strongly invariant across hold-out validation samples ($\Delta\chi^2(80) = 79.98$, $p = .48$). Reliability estimates ranged from $\omega = .86$ to $.91$.

Dysregulation is defined as a construct with specific domains [39, 64] and thus assumed to be best modeled in a bifactor S-1 model with a directly measured G factor [65]. The model showed good fit to the data. When controlling for general dysregulation, the correlation of the domains (specific factors) were as follows: behavioral and cognitive dysregulation ($r = .30$ [.21 - .39]), behavioral and emotional dysregulation ($r = .53$ [.46 - .60]), cognitive and emotional dysregulation ($r = .62$ [.56 - .69]).

**Avoidance behavior.** For avoidance behavior, we decided to select four items for each of the two subscales (8 of 14 items in total). A hold-out validation brute force algorithm was run which yielded a robust solution that was strongly invariant across calibration and validation sample ($\Delta\chi^2(40) = 42.23$, $p = .37$). The measurement model demonstrated excellent fit to the data. Reliability estimates for the two scales were $\omega = .72$ and $\omega = .93$.

**Clinical significance.** Twelve items (out of 19) were selected through a hold-out validation ACO algorithm with a function additionally minimizing the latent correlation. The model demonstrated good fit to the data and was strongly invariant across the calibration and validation sample ($\Delta\chi^2(60) = 74.22$, $p = .10$). The correlation of the subscales was still very high ($r = .91$), however, the selected model was the best possible solution. Reliability estimates for the two scales were $\omega = .94$ and $\omega = .92$.

**Joint model.** We analyzed a joint model incorporating fifteen of the twenty symptom-oriented scales. Reactions to the anticipation of sounds were excluded since they are subordinate in the measurement of misophonia. The joint model demonstrated good fit to the data, which is an evidence for the distinctiveness of the symptom-oriented scales. An overview of the descriptive and psychometric properties of the final selected items for the BMQ-R are given in S1 Table. To analyze the dimensionality of highly correlated subscales ($r > .80$), for each combination of highly correlated scales, we compared a one-factor model with a two-factor correlated model via likelihood-ratio tests. The results of these tests can be found in S2 Table.

## Construct validity

To evaluate construct validity beyond the factor structure, we calculated latent intercorrelations between the symptom-oriented scales and the GSIS scale, which are given in Table 4, as

**Table 4. Latent intercorrelations and reliability estimates for the BMQ-R symptom part and general sound intolerance symptoms.**

| Measure | 1 | 2 | 3 | 4 | 5 | 6 | 7 | 8 | 9 | 10 | 11 | 12 | 13 | 14 | 15 | 16 | 17 | 18 | 19 | 20 | 21 |
|---|---|---|---|---|---|---|---|---|---|---|---|---|---|---|---|---|---|---|---|---|---|
| 1. Anger Pres. | (.90) | - | - | - | - | - | - | - | - | - | - | - | - | - | - | - | - | - | - | - | - |
| 2. Irritation Pres. | **.88** | (.77) | - | - | - | - | - | - | - | - | - | - | - | - | - | - | - | - | - | - | - |
| 3. Disgust Pres. | .55 | .65 | (.91) | - | - | - | - | - | - | - | - | - | - | - | - | - | - | - | - | - | - |
| 4. Anxiety Pres. | .60 | .73 | .49 | (.88) | - | - | - | - | - | - | - | - | - | - | - | - | - | - | - | - | - |
| 5. Physical Pres. | .80 | **.86** | .55 | **.83** | (.85) | - | - | - | - | - | - | - | - | - | - | - | - | - | - | - | - |
| 6. Anger Ant. | .75 | .62 | .46 | .48 | .59 | (.93) | - | - | - | - | - | - | - | - | - | - | - | - | - | - | - |
| 7. Irritation Ant. | .52 | .56 | .41 | .48 | .53 | **.83** | (.87) | - | - | - | - | - | - | - | - | - | - | - | - | - | - |
| 8. Disgust Ant. | .41 | .47 | **.89** | .44 | .48 | .58 | .59 | (.87) | - | - | - | - | - | - | - | - | - | - | - | - | - |
| 9. Anxiety Ant. | .59 | .69 | .39 | **.87** | .73 | .57 | .62 | .46 | (.90) | - | - | - | - | - | - | - | - | - | - | - | - |
| 10. Physical Ant. | .66 | .69 | .50 | .71 | **.85** | **.84** | **.86** | .67 | .80 | (.84) | - | - | - | - | - | - | - | - | - | - | - |
| 11. Disprop. | .67 | .64 | .39 | .39 | .62 | .47 | .36 | .31 | .44 | .47 | (.90) | - | - | - | - | - | - | - | - | - | - |
| 12. Excess | **.83** | **.83** | .49 | .65 | **.81** | .63 | .54 | .43 | .70 | .68 | **.83** | (.91) | - | - | - | - | - | - | - | - | - |
| 13. Gen. Dys. | .70 | .69 | .38 | .46 | .62 | .54 | .41 | .32 | .51 | .55 | .57 | .70 | (.91) | - | - | - | - | - | - | - | - |
| 14. Behav. Dys. | .80 | .71 | .48 | .51 | .69 | .64 | .47 | .40 | .51 | .58 | .66 | .75 | .68 | (.86) | - | - | - | - | - | - | - |
| 15. Cogn. Dys. | .78 | **.84** | .52 | .56 | .74 | .58 | .52 | .44 | .59 | .62 | .62 | .79 | .71 | .63 | (.90) | - | - | - | - | - | - |
| 16. Emot. Dys | **.83** | **.88** | .52 | .73 | **.87** | .64 | .58 | .45 | .74 | .73 | .67 | **.94** | .70 | .76 | .81 | (.87) | - | - | - | - | - |
| 17. React. Avoid. | .71 | .79 | .57 | .65 | .79 | .55 | .52 | .53 | .67 | .69 | .56 | .72 | .62 | .63 | .75 | .79 | (.72) | - | - | - | - |
| 18. Ant. Avoid. | .59 | .60 | .40 | .59 | .63 | .46 | .44 | .37 | .68 | .61 | .41 | .57 | .49 | .50 | .62 | .67 | **.83** | (.93) | - | - | - |
| 19. Distress | **.81** | **.85** | .49 | .74 | **.86** | .71 | .61 | .44 | .78 | .78 | .65 | **.86** | .66 | .74 | .75 | **.90** | .76 | .69 | (.94) | - | - |
| 20. Funct. Imp. | .72 | .79 | .45 | .72 | .78 | .62 | .55 | .45 | .78 | .72 | .57 | .78 | .60 | .69 | .69 | **.81** | .74 | .73 | **.92** | (.92) | - |
| 21. GSIS | **.85** | **.93** | .53 | .56 | .79 | .65 | .53 | .41 | .56 | .64 | .65 | .77 | .67 | .73 | .79 | **.83** | .78 | .63 | **.84** | .75 | (.84) |

*N* = 611–951 for grey shaded cells, which represent intercorrelations within symptom areas, and reliability estimates. All intercorrelations across symptom areas were estimated based on *N* = 589–616. Pres. = Presence; Ant. = Anticipation; Disprop. = Disproportionality; Gen. Dys. = General Dysregulation; Behav. Dys. = Behavioral Dysregulation; Cogn. Dys. = Cognitive Dysregulation; Emot. Dys. = Emotional Dysregulation; React. Avoid. = Reactive Avoidance; Ant. Avoid. = Anticipatory Avoidance; Funct. Imp. = Functional Impairment; GSIS = General Sound Intolerance Symptoms. Dysregulation is herein specified in a correlated first-order CFA instead of a bifactor S-1 model. McDonald's ω based on the respective confirmatory factor analyses are in parentheses on the diagonal. Correlations > .80 are in bold. All correlations were significant at *p* < .001.

well as specific hypothesis-driven correlation matrices within the scales' specific content areas. Effect sizes were interpreted in line with Cohen [66].

**Aversive reactions to the presence or anticipation of sounds.** As hypothesized, present anger and irritation reactions displayed a strong positive correlation (*r* = .88) as well as present irritation with physical reactions (*r* = .86). Moreover, present physical reactions displayed strong positive correlations with each misophonic emotional reaction to the presence of sounds (*r* = .55 - .86). A similar pattern could be observed for anticipatory reactions: irritation correlated to *r* = .83 with anger and to *r* = .86 with physical reactions. For anticipatory physical reactions, high correlations with each misophonic emotional reaction to the anticipation of sounds were observed (*r* = .67 - .86). Regarding intercorrelations between present and anticipatory misophonic reactions, the same emotional reactions correlated higher than different emotional reactions. Anticipatory physical reactions were equally correlated with present physical reaction compared to anticipatory anger and irritation reactions.

To further scrutinize the construct validity of emotional and physical misophonic reactions, convergent measures were correlated, respectively. For anger reactions, convergent anger, verbal aggression, physical aggression, and hostility were assessed with the AQ. The first-order factor model had to be optimized through a hold-out validation ACO algorithm. Therefore, 21 out of the 27 items were selected to achieve acceptable fit to the data. Reliability estimates were

**Table 5. Latent intercorrelations for misophonic anger and irritation reactions with convergent measures.**

| Measure | 1 | 2 | 3 | 4 | 5 | 6 | 7 | 8 | 9 | 10 | 11 |
|---|---|---|---|---|---|---|---|---|---|---|---|
| 1. BMQ-R: Anger Presence | (.90) | - | - | - | - | - | - | - | - | - | - |
| 2. BMQ-R: Anger Anticipation | .75 | (.93) | - | - | - | - | - | - | - | - | - |
| 3. S-Five-E: External Appraisals | .58 | .55 | (.92) | - | - | - | - | - | - | - | - |
| 4. S-Five-E: Outbursts | .74 | .64 | .54 | (.87) | - | - | - | - | - | - | - |
| 5. AQ: Anger | .58 | .56 | .46 | .67 | (.82) | - | - | - | - | - | - |
| 6. AQ: Verbal Aggression | .37 | .36 | .39 | .46 | .78 | (.63) | - | - | - | - | - |
| 7. AQ: Physical Aggression | .28 | .31 | .30 | .60 | .54 | .58 | (.71) | - | - | - | - |
| 8. AQ: Hostility | .35 | .43 | .40 | .43 | .70 | .70 | .45 | (.79) | - | - | - |
| 9. BMQ-R: Irritation Presence | .88 | .62 | ..60 | .59 | .50 | .36 | .19 | .37 | (.77) | - | - |
| 10. BMQ-R: Irritation Anticipation | .52 | .83 | .45 | .50 | .46 | .28 | .21 | .43 | .56 | (.87) | - |
| 11. BITe: Irritation | .62 | .57 | .48 | .57 | .72 | .48 | .37 | .62 | .62 | .49 | (.91) |

$N$ = 580–639 BMQ-R = Berlin Misophonia Questionnaire Revised; S-Five-E = Selective Sound Sensitivity Syndrome Scale Experiences; AQ = Aggression Questionnaire; BITe = Brief Irritability Test. McDonald's ω based on the confirmatory factor analyses of the respective scales are in parentheses on the diagonal. Intercorrelations of the BMQ-R subscales are adopted from Table 4 and hence $N$ = 588–616.

All correlations were significant at $p < .001$.

still similar to the original scale. Further, two scales from the S-Five-E were used (externalising appraisals and outbursts). The complete S-Five-E model showed acceptable model fit compared to the pretest. With respect to irritation, the BITe was used as a convergent measure.

Latent intercorrelations for anger and irritation reactions with convergent measures are given in Table 5. Present and anticipatory misophonic anger correlated to $r = .74$ respectively $r = .64$ with misophonic outbursts whereas present and anticipatory irritation correlated to $r = .59$ respectively $r = .50$ with misophonic outbursts. Externalising appraisals correlated marginally lower with irritation and anger reactions ($r = .45 - .60$). Regarding convergent anger, correlations with present and anticipatory anger were $r = .56-.58$ and with irritation $r = .46 -.50$. In contrast, present and anticipatory irritation correlated with convergent irritation to $r = .62$ respectively $r = .49$, whereas present and anticipatory anger correlated to $r = .62$ respectively $r = .57$. Furthermore, verbal aggression, physical aggression, and hostility correlated moderately with present and anticipatory anger ($r = .28 - .43$). For anticipated misophonic irritation a similar pattern could be observed ($r = .21 - .43$). On average, present anger was only marginally higher correlated with convergent anger and aggression measures than present irritation ($\bar{r} = .40$ vs. $\bar{r} = .36$) and anticipated anger correlated higher than anticipated irritation ($\bar{r} = .42$ vs. $\bar{r} = .35$).

To investigate the construct validity of the BMQ-R disgust reactions, the disgust propensity scale from the DPSS-R was used. Disgust propensity correlated highly positively with misophonic disgust reactions to the presence of sounds ($r = .72$) and to the anticipation of sounds ($r = .68$) which is higher than the average cross-emotional intercorrelation of misophonic disgust scales ($\bar{r} = .52$). Anticipatory disgust and present disgust correlated to $r = .88$ [.85 - .92].

For misophonic anxiety reactions, cognitive and somatic anxiety symptoms measured with the STICSA as well as threat measured by the S-Five-E were used as convergent constructs. Furthermore, physical misophonic symptoms were correlated with somatic anxiety since somatic anxiety symptoms measured by the STICSA are not necessarily specific to anxiety. The STICSA 18-item model from the pretest demonstrated good fit to the data. Latent intercorrelations for misophonic anxiety and physical reactions with convergent measures are given in Table 6. Present and anticipatory misophonic anxiety correlated strongly with

**Table 6. Latent intercorrelations for misophonic anxiety and physical reactions with convergent measures.**

| Measure | 1 | 2 | 3 | 4 | 5 | 6 | 7 |
|---|---|---|---|---|---|---|---|
| 1. BMQ-R: Anxiety Presence | (.85) | - | - | - | - | - | - |
| 2. BMQ-R: Anxiety Anticipation | .87 | (.90) | - | - | - | - | - |
| 3. S-Five-E: Threat | .77 | .80 | (.92) | - | - | - | - |
| 4. STICSA-T: Cognitive Anxiety Symptoms | .61 | .66 | .67 | (.91) | - | - | - |
| 5. STICSA-T: Somatic Anxiety Symptoms | .73 | .70 | .70 | .69 | (.90) | - | - |
| 6. BMQ-R: Physical Presence | .83 | .80 | .77 | .62 | .89 | (.85) | - |
| 7. BMQ-R: Physical Anticipation | .71 | .73 | .73 | .63 | .78 | .85 | (.84) |

$N$ = 567–639. BMQ-R = Berlin Misophonia Questionnaire Revised; STICSA-T = State-Trait Inventory for Cognitive and Somatic Anxiety—Trait Scales. McDonald's ω based on the confirmatory factor analyses of the respective scales are in parentheses on the diagonal. Intercorrelations of the BMQ-R subscales are adopted from Table 4 with $N$ = 588–616.

All correlations were significant at $p < .001$.

perceived threat ($r = .776$ and $r = .80$) as well as with somatic anxiety symptoms ($r = .73$ and $r = .70$) and cognitive anxiety symptoms ($r = .61$ and $r = .66$). On the other hand, physical reactions to the presence or anticipation of sounds were higher correlated with somatic anxiety ($r = .89$ and $r = .78$) than with cognitive anxiety ($r = .62$ and $r = .63$). Further, present physical reactions and present anxiety were similarly equally strongly correlated with perceived threat compared to present anxiety (both $r = .77$ respectively $r = .76$) whereas anticipatory physical reactions had a lower correlation ($r = .73$) than anticipatory anxiety ($r = .80$). Present and anticipatory anxiety correlated to $r = .875$ [.84 - .90].

**Clinical insight.** To test convergent validity of the two clinical insight scales, correlations with internal appraisals and external appraisals from the S-Five-E as well as the illness coherence scale from the IPQ-MH were calculated. Latent intercorrelations for recognition of disproportionality and excess with convergent measures are given in Table 7. Contrary to our hypotheses, illness coherence was moderately to highly negatively correlated with recognition of disproportionality and excess ($r = -.50$ respectively $r = -.45$). On the other hand, the scale internal appraisals (symptom attribution) displayed strong positive correlations with both recognition of disproportionality ($r = .65$) and excess ($r = .70$) whereas external appraisals were moderately correlated with recognition of disproportionality ($r = .33$) and excess ($r = .47$).

Furthermore, high average correlations with dysregulation symptoms measured by the BMQ-R were observed for recognition of disproportionality ($\bar{r} = .64$) and for recognition of excess ($\bar{r} = .79$). Also, distress and functional impairment as measured by the BMQ-R were

**Table 7. Latent intercorrelations for recognition of excess and disproportionality with convergent measures.**

| Measure | 1 | 2 | 3 | 4 | 5 |
|---|---|---|---|---|---|
| 1. BMQ-R: Disproportionality | (.90) | - | - | - | - |
| 2. BMQ-R: Excess | .83 | (.91) | - | - | - |
| 3. IPQ-MH: Illness Coherence | -.50 | -.45 | (.92) | - | - |
| 4. S-Five-E: Internal Appraisals | .65 | .70 | -.41 | (.93) | - |
| 5. S-Five-E: External Appraisals | .33 | .47 | -.17 | .41 | (.92) |

$N$ = 553–639. BMQ-R = Berlin Misophonia Questionnaire Revised; IPQ-MH = Illness Perception Questionnaire Mental Health; S-Five-E = Selective Sound Sensitivity Syndrome Scale Experiences. McDonald's ω based on the confirmatory factor analyses of the respective scales are in parentheses on the diagonal. Intercorrelations of the BMQ-R subscales are adopted from Table 4 with $N$ = 696.

All correlations were significant at $p < .001$.

strongly correlated with both clinical insight dimensions ($r$ = .57 - .86). Recognition of disproportionality and excess were strongly correlated with overall symptom burden measured by the MisoQuest ($r$ = .76 respectively $r$ = .88) as well as moderately to highly correlated with impact measured by the S-Five-E ($r$ = .45 respectively $r$ = .62).

**Dysregulation.** To validate misophonic dysregulation defined as a general construct with specific subfacets, scales from the DERS were chosen. More specifically, items from the scales impulse control difficulties, difficulties engaging in goal-oriented behavior, and limited access to emotion regulation were selected to measure behavioral, cognitive, and emotional dysregulation, respectively. One item comprises general dysregulation with respect to the content, which led to specifying an S•I-1 model [65] with this item as the reference factor and the other items as specific factors regarding their dysregulation facet. To compare the results, a correlated first-order factor model without the general item (factor) was also specified. Both models demonstrated good fit to the data. For latent intercorrelations between these scales see Table 8. For the correlated first-order models, medium to strong correlations between general misophonic dysregulation and its facets with convergent measures of dysregulation were observed ($r$ = .45 - .65, $\bar{r}$ = .56). Behavioral dysregulation correlated highest with difficulties in impulse control ($r$ = .62) and cognitive dysregulation correlated highest with difficulties engaging in goal-directed behavior. This clear pattern could not be observed for emotional dysregulation being highest correlated with general control difficulties and similarly with other DERS facets.

General control difficulties as defined in the S•I-1 model correlated highly with general dysregulation. However, their correlation with impulse control difficulties was higher than with general control difficulties.

Furthermore, outbursts as measured by the S-Five-E were strongly correlated with misophonic dysregulation. General dysregulation was correlated to $r$ = .62 with outbursts. Regarding the dysregulation domains, outbursts were correlated to $r$ = .83 with behavioral, $r$ = .55 with cognitive and $r$ = .64 with emotional dysregulation.

**Avoidance behavior.** For reactive and anticipatory avoidance behavior, the NAQ and the BEAQ (Behavioral Avoidance) scales were chosen to test convergent validity. Regarding the NAQ, a brute force hold-out optimization was applied since the unidimensional factor model demonstrated poor fit to the data. Eleven items were selected which led to a good model fit.

**Table 8. Latent intercorrelations for misophonic dysregulation with convergent measures.**

| Measure | 1 | 2 | 3 | 4 | 5 | 6 | 7 | 8 |
|---|---|---|---|---|---|---|---|---|
| 1. BMQ-R: General Dysregulation | (.91) | - | - | - | - | - | - | - |
| 2. BMQ-R: Behavioral Dysregulation | .68 | (.86) | - | - | - | - | - | - |
| 3. BMQ-R: Cognitive Dysregulation | .71 | .63 | (.87) | - | - | - | - | - |
| 4. BMQ-R: Emotional Dysregulation | .70 | .76 | .81 | (.87) | - | - | - | - |
| 5. DERS: General Control Difficulties | .54 | .58 | .57 | .70 | (.83)[a] | - | - | - |
| 6. DERS: Impulse Control Difficulties | .62 | .62 | .53 | .61 | .00[z] | (.70) | - | - |
| 7. DERS: Difficulties Engaging in Goal-Oriented Behavior | .49 | .45 | .64 | .65 | .00[z] | .73 | (.90) | - |
| 8. DERS: Limited Access to Emotion Regulation | .47 | .48 | .57 | .62 | .00[z] | .76 | .84 | (.91) |

$N$ = 570–587. BMQ-R = Berlin Misophonia Questionnaire Revised; DERS = Difficulties in Emotion Regulation Scale. Correlations are based on first-order factor analyses, except for correlations of the DERS: General Control Difficulties factor for which the bifactor S•I-1 model was used. McDonald's ω based on the first-order confirmatory factor analyses of the respective scales are in parentheses on the diagonal. Intercorrelations of the BMQ-R subscales are adopted from Table 4 and hence $N$ = 708.

[a] McDonald's ω as defined in the bifactor S•I-1 model.

[z] Correlations are per definition set to zero.

All correlations were significant at $p < .001$.

With respect to the content, we carefully considered minimizing the loss of content through item reduction. Internal consistency was similar to the estimate from the development study of the NAQ ($\omega = .91$). The BEAQ subscale demonstrated good model fit and reliability ($\omega = .85$).

Avoidance of daily sounds as measured by the NAQ correlated strongly with both reactive misophonic avoidance ($r = .64$) and anticipatory misophonic avoidance ($r = .74$). Further, reactive and anticipatory avoidance were strongly correlated with behavioral avoidance ($r = .57$ respectively $r = .62$). We also tested the relationship between avoidance behavior and impeded avoidance as measured by the perceived threat scale of the S-Five-E which was strong for both reactive avoidance ($r = .76$) and anticipatory avoidance ($r = .69$).

Further evidence on construct validity could be demonstrated through a high average correlation of reactive avoidance with present aversive emotional and physical misophonic reactions measured by the BMQ-R ($\bar{r} = .71$) which was higher than the average correlation of anticipatory avoidance ($\bar{r} = .59$).

**Clinical significance.** Construct validity for misophonic distress and functional impairment was assessed by correlations with different dimensions of functional impairment measured by the WHODAS 2.0 as well as overall symptom burdening as measured by the MisoQuest and misophonic impact measured by the S-Five-E. For latent correlations between misophonic distress and functional impairment with convergent measures see Table 9. Correlations with WHODAS 2.0 scales were calculated based on maximum likelihood robust estimation with full information likelihood to address the substantial number of missing values for these scales. Misophonic impact correlated strongly with distress ($r = .76$) and functional impairment ($r = .84$). Overall symptom burden as measured by the MisoQuest was strongly correlated with both distress ($r = .90$) and functional impairment ($r = .85$). With respect to impairment domains by the WHODAS 2.0, functional impairment was most strongly correlated with impairment in participation in society ($r = .75$) and impairment in getting along with people ($r = .61$), whereas impairment in household activities was moderately correlated ($r = .44$). Distress demonstrated a similar pattern, although on average being less highly correlated. More specifically, impairment in participation in society is correlated to $r = .64$, impairment in getting along with people to $r = .53$ and impairment in household activities to $r = .37$.

**Table 9. Latent intercorrelations for misophonic distress and functional impairment with convergent measures.**

| Measure | 1 | 2 | 3 | 4 | 5 | 6 | 7 | 8 | 9 |
|---|---|---|---|---|---|---|---|---|---|
| 1. BMQ-R: Distress | (.94) | - | - | - | - | - | - | - | - |
| 2. BMQ-R: Functional Impairment | .91 | (.95) | - | - | - | - | - | - | - |
| 3. S-Five-E: Impact | .76 | .84 | (.90) | - | - | - | - | - | - |
| 4. WHODAS 2.0: Cognition | .48 | .51 | .54 | (.86) | - | - | - | - | - |
| 5. WHODAS 2.0: Getting Along with People | .53 | .61 | .66 | .81 | (.82) | - | - | - | - |
| 6. WHODAS 2.0: Household Activities | .37 | .44 | .43 | .66 | .71 | (.96) | - | - | - |
| 7. WHODAS 2.0: Work or School Activities | .45 | .52 | .58 | .71 | .72 | .63 | (.93) | - | - |
| 8. WHODAS 2.0: Participation | .64 | .75 | .77 | .71 | .84 | .68 | .74 | (.91) | - |
| 9. MisoQuest | .90 | .85 | .73 | .54 | .56 | .36 | .46 | .63 | (.93) |

*N* = 585–620. BMQ-R = Berlin Misophonia Questionnaire Revised; S-Five-E = Selective Sound Sensitivity Syndrome Scale Experiences; WHODAS 2.0 = World Health Organization Disability Assessment Schedule 2.0. McDonald's ω based on the confirmatory factor analyses of the respective scales are in parentheses on the diagonal. All correlations were significant at $p < .001$.

## Discussion

This study described the optimization of the BMQ and reported an extensive examination of the new BMQ-R's psychometric properties through structural analyses, reliability assessment and correlative analyses for the purpose of testing construct validity. We presented a symptom-oriented modeling approach for the measurement of the diagnostic criteria of misophonia proposed by Jager et al. [1], which enables a precise and comprehensive assessment of misophonic symptoms. Within this framework, previous items from the first version as well as newly constructed items were tested in a large sample comprising affected as well as non-affected individuals. An algorithmic hold-out validation item selection procedure enabled the formation of 20 symptom-oriented scales which encompass 73 items in six criteria-oriented content areas: (1) aversive reactions to the presence and (2) anticipation of sounds, (3) clinical insight, (4) dysregulation, (5) avoidance behavior, and (6) clinical significance. About 38% of the original items remained in the final BMQ-R.

With respect to the factorial validity, all developed symptom-oriented models showed good to excellent fit to the data. Accordingly, the models demonstrate a precise assessment and distinction of misophonic symptoms. A joint model comprising the essential 15 symptom scales is a preliminary evidence of the robustness and clarity of the factorial structure. Still, misspecifications of the model are evident for some items. Further developments of the questionnaire should particularly scrutinize these issues. Nevertheless, these findings show evidence for the overall structure of the BMQ-R and represent an improvement compared to the first version. Additional analyses on the dimensionality of highly correlated scales as shown in the table in S2 Table lead to the same conclusion.

The assessment of the reliability demonstrated good psychometric properties of the developed scales. Estimates of ω ranged from .82 to .95 except for irritation reactions to the presence of sounds and reactive avoidance behavior (ω = .77 and ω = .72).

To assess construct validity, several correlative hypotheses regarding convergent measures of the respective scales were investigated, and results can be interpreted in a nomological network. One of the central aims of the validation was the distinction between different aversive misophonic reactions as reported in the literature [1, 6–8]. Because anger and irritation are related, yet different constructs [23], we aimed to distinguish these emotional reactions in the BMQ-R. Results demonstrated partial evidence for this distinction through stronger associations of anger reactions with misophonic outbursts, convergent anger, and physical aggression. Contrary to our predictions, misophonic anger correlated as highly with a convergent irritation measure as misophonic irritation. Hence, our findings do not support a clear distinction between these constructs and raise questions about the theoretical distinction between irritation and anger in the measurement of misophonia. However, associations between irritation and other scales of the BMQ-R demonstrate that irritation is uniquely related to other misophonic symptoms compared to anger reactions (e.g., lower correlations with behavioral dysregulation and recognition of disproportionality as well as a higher correlation with the GSIS scale). We therefore argue that irritation reactions need to be further investigated in future studies but appear to be an important aspect of misophonic experiences. Dimensionality analyses also support the distinction between anger and irritation in the measurement of misophonia (see the table in S2 Table).

Regarding disgust, a clear convergent correlation with disgust propensity demonstrates evidence for the construct validity. Anticipatory disgust, however, was very strongly associated with present disgust ($r = .89$) and the scales were on average similarly correlated with other misophonic symptom scales ($\bar{r} = .55$ for present disgust and $\bar{r} = .41$ for anticipatory disgust). This result indicates construct validity but also shows that the constructs are not clearly

distinct from each other. The analysis of the two scales' dimensionality also provides evidence for the distinction of anticipatory and present disgust (see S2 Table).

For anxiety reactions, all validation hypotheses were corroborated. While Jager et al. [1] considered present anxiety reactions as subordinate, our results indicate that present anxiety reactions are substantially correlated to emotional dysregulation, avoidance, distress and functional impairment, rendering it an important dimension of misophonic experience. Present anxiety reactions were also moderately to strongly associated with other aversive misophonic reactions which emphasizes the importance of measuring this dimension. However, the high intercorrelation of present and anticipatory anxiety ($r = .87$) raises the question whether present anxiety truly measures present anxiety-related thoughts and not anticipatory thoughts in the presence of sounds. This doubt is reinforced by the fact that the average intercorrelations with other misophonic symptoms apart from the aversive emotional and physical reactions were similar ($\bar{r} = .39$ for present anxiety and $\bar{r} = .44$ for anticipatory anxiety). We therefore recommend further investigating the relationship between present and anticipatory anxiety focusing on the discriminant validity of these scales with respect to the source of anxious thoughts (future misophonic situations vs. perceived threat due to the sound itself). Nevertheless, we favor the usage of both dimensions since content-related validity respectively item wording suggests a sufficient distinction and dimensionality analyses also support this distinction.

Physical misophonic reactions displayed strong positive correlations which were substantially higher with somatic anxiety symptoms than misophonic anxiety reactions. This clearly supports the convergent validity of physical reactions. We also observed strong positive associations of both present and anticipatory physical reactions with each emotional misophonic response. Because the physical reaction is not specific to any emotional response but largely caused by it, high correlations were expected and hence support the construct validity. Likewise, we expected a large association between present and anticipatory physical reactions, but the observed correlation was very high ($r = .85$), casting doubt on the scales' discriminant validity. The analysis of the two scales' dimensionality also supports a distinction of anticipatory and present physical reactions.

The outcomes are heterogeneous for clinical insight. On the one hand, dimensions of clinical insight correlated strongly with internal appraisals of symptoms and moderately with external appraisals, which demonstrates a clear convergent correlative pattern. Individuals who tend to internally attribute their symptoms are more likely to recognize excessive and disproportionate behavior. For external attributions, this relationship was weaker, which is in line with theoretical considerations: Individuals recognize that their misophonic reactions are strong but attribute their reactions to others' bad-mannered behavior. Contrary to our hypotheses, illness coherence was highly negatively instead of positively correlated with clinical insight dimensions. Hence, individuals less comprehending of their symptoms are more likely to recognize excessive behavior and disproportionality or vice versa. This might be explained by the fact that illness coherence correlates strongly positively with personal control over symptoms [28, 67]. Individuals less comprehending of their symptoms could hence be more likely to perceive less personal control over their symptoms. Our data also support this relationship since misophonic dysregulation facets (as an expression of loss of control) display a moderate negative correlation with illness coherence ($r = -.32$ to $-.36$). On that account, the observed negative relationship between illness coherence and recognition of excess and disproportionality supports the convergent validity of these scales, albeit not expected. A further examination of the construct validity of the clinical insight dimensions is necessary. However, to our knowledge, no suitable insight scale directly measuring the intended insight dimensions exists.

Modeling misophonic dysregulation through the inclusion of specific domains (behavioral, cognitive, and emotional dysregulation) allows for an extensive measurement of misophonic dysregulation and loss of control. The specified correlated first-order model demonstrated the adequacy of such a distinction for misophonic symptoms and simultaneously incorporated contemporary findings on misophonia from the recent literature. Behavioral and cognitive dysregulation correlated more strongly with their corresponding convergent measure than other domains, which supports the validity of their distinction. For emotional dysregulation, however, the predicted pattern did not emerge. This is reflected through stronger associations between emotional dysregulation and general control difficulties than limited access to emotion regulation. Emotional dysregulation is also similarly associated with the subfacets of the convergent DERS scale. These results raise the question whether emotional dysregulation is sufficiently captured by the BMQ-R. Regarding the content, the items mainly reflect intense emotional experience in the sense of loss of control over emotions which was exactly the intention. Future studies should consider alternative validation measures of emotional dysregulation that emphasize a loss of control over emotional experience to assess construct validity. Another piece of evidence on construct validity were strong associations with misophonic outbursts which comprise verbally and physically aggressive outbursts. As predicted, behavioral dysregulation correlated most strongly with outbursts. A more general criticism of the definition of misophonic dysregulation is that it might be too broad and generic to capture the entire construct of misophonic dysregulation, even though different dysregulation domains have been incorporated. Our treatment of dysregulation domains is a condensed derivation of dysregulation dimensions described by D'Agostino et al. [64] and Gratz and Roemer [39] as well as misophonia-specific findings [19]. Dimensions that are not or only marginally included are 'decreased emotional awareness' and 'cognitive reappraisal difficulty' [64] as well as 'non-acceptance of emotions' [39]. Especially the last two dimensions might be relevant in the assessment of misophonic dysregulation. We therefore advocate the examination of misophonic dysregulation in further studies. Nonetheless, the present evidence on the scales' construct validity indicates that the distinction of behavioral, cognitive, and emotional domains is a useful and valid approach to quantify misophonic dysregulation.

The validity of misophonic avoidance behavior could be shown clearly through high correlations with convergent sound avoidance and behavioral experiential avoidance. Since the NAQ largely comprises items assessing the anticipatory nature of sound avoidance, it is not surprising that we found higher correlations with anticipatory avoidance than with reactive avoidance. Future research should focus on validating the scale reactive avoidance by measuring escape behavior more directly.

For misophonic distress and functional impairment, several strong correlations with convergent measures were observed. It could be shown that both functional impairment and distress were strongly related to misophonic impact, impairments in the participation of society and impairments in getting along with people. Also, overall symptom burden was strongly correlated with the two dimensions of clinical significance. The findings clearly support the convergent validity of clinical significance dimensions.

Beside the outlined improvements of the symptom part, the condition part, which comprises the GSIS scale and sound disturbance of specific sound classes, was also extended and revised. The GSIS model showed good fit to the data and good internal consistency. GSIS expectedly correlated moderately to strongly with other misophonic symptoms. However, the large association between GSIS and present misophonic irritation ($r = .93$) questions the incremental utility of such a screening scale. However, the average item difficulty of GSIS ($\bar{P}_i = 90.20$) compared to present irritation reactions ($\bar{P}_i = 76.37$) supports the usefulness of both

scales. Additional analyses on the dimensionality of both scales support the distinction between GSIS and irritation (see S2 Table). The usage of the GSIS scale as a screening prior to the application of the BMQ-R symptoms part is hence justifiable through considerations regarding measurement theory as well as item content.

## Limitations

There are several limitations of the study that should be considered when interpreting the results. Because our sample was mainly drawn from social media groups, the study is not representative for the German population. Unfortunately, the sample only had a small proportion of male participants (12.6%), which impairs the comparability of results regarding gender. Further studies should therefore test the BMQ-R in a large representative sample. Additionally, an extensive examination of measurement invariance (especially across gender, age, and sound classes) is indicated. The study length ($Mdn$ = 43 minutes) might have decreased data quality due to exhaustion effects, although the data quality examination should have minimalized this effect.

From a methodological perspective, the applied probabilistic ACO item selection procedure might have been extended by using multiple runs with different criteria to ensure detecting a global best solution for the specific models. Beyond that, all measures are self-report scales sharing method specific variance that artificially increases correlations of the constructs [68]. An extension in terms of multitrait-multimethod analyses is thus called for.

For future research, it is important to replicate the factor structure in a more representative sample. An extended validation study especially investigating discriminant, differential and content validity is required. For the clinical or individual application of the BMQ-R, a large representative norming sample should be drawn. However, this study's large sample already provides initial comparative values for clinical practice. A clinical cutoff must be determined to analyze the sensitivity and specificity of the instrument and to classify affected individuals. To date, there is no instrument that measures temporal and trans-situational stability as well as situational influences of misophonic symptoms. The latent state-trait theory provides an approach to develop a state-trait instrument [69] which certainly would be a scientifically beneficial extension of the BMQ-R.

## Conclusions

Several limitations of previous and recent instruments [15–17] have been addressed in the development of the BMQ-R. Previous instruments measuring misophonia neglect important aspects of the condition which are now incorporated. These aspects include the distinction between behavioral, cognitive, and emotional dysregulation; impulsive physical reactions and clinical insight; as well as the discrimination of reactive and anticipating avoidance strategies. A thorough and comprehensive assessment of misophonic emotional responses is unique to the BMQ-R. The present study demonstrates that the BMQ-R allows for a reliable and valid measurement of the symptoms and therefore enables further research on misophonia on different levels. Through the symptom-oriented modeling approach, a clearer measurement of the criteria is viable, with the option to regard accessory symptoms. The presented evidence on construct validity elucidates the multidimensional nature of self-reported misophonic symptoms and demonstrates the validity of the diagnostic criteria by Jager et al. [1]. Furthermore, the BMQ-R extends several diagnostic criteria. This allows for both classifying individuals according to the criteria and assessing the wide range of misophonic symptoms in line with the scientific consensus [8]. The symptom-oriented approach also enables researchers to investigate specific symptoms or scales of interest apart from the others and to scrutinize

relationships to other symptoms or traits in detail. Conclusively, the BMQ-R fills a major research gap in measuring misophonia which makes it possible to investigate causes and treatments of this severe condition.

## Supporting information

**S1 Text. Pretest of the Berlin Misophonia Questionnaire (BMQ) main study.**
(DOCX)

**S1 Table. Descriptive statistics and psychometric properties of the BMQ-R items.**
(DOCX)

**S2 Table. Analyses of the dimensionality of highly correlated BMQ-R symptom-scales.**
(DOCX)

## Acknowledgments

We thank Kevin Sturm, Maik Hante, Rebecca Gruzman, Gloria Gierlach, Friedemann Trutzenberg, and Nicole Chantal Skerstupeit for their translations of the validation scales; and Taym Asalti for helpful comments and suggestions on our manuscript.

## Author Contributions

**Conceptualization:** Nico Remmert, Katharina Maria Beate Schmidt, Patrick Mussel.

**Data curation:** Nico Remmert.

**Formal analysis:** Nico Remmert, Minne Luise Hagel.

**Investigation:** Nico Remmert, Katharina Maria Beate Schmidt.

**Methodology:** Nico Remmert, Patrick Mussel, Michael Eid.

**Project administration:** Nico Remmert.

**Software:** Minne Luise Hagel, Michael Eid.

**Supervision:** Katharina Maria Beate Schmidt, Patrick Mussel, Michael Eid.

**Validation:** Nico Remmert, Patrick Mussel, Minne Luise Hagel, Michael Eid.

**Visualization:** Nico Remmert.

**Writing – original draft:** Nico Remmert.

**Writing – review & editing:** Nico Remmert, Katharina Maria Beate Schmidt, Patrick Mussel, Minne Luise Hagel, Michael Eid.

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
