## [Decision Letter · Decision Letter 0]

14 Mar 2022

PONE-D-21-25396

The Berlin Misophonia Questionnaire (BMQ): Development and validation of a symptom-oriented diagnostical instrument for the measurement of misophonia

PLOS ONE

Dear Dr. Remmert,

Thank you for submitting your manuscript to PLOS ONE. After careful consideration, we feel that it has merit but does not fully meet PLOS ONE’s publication criteria as it currently stands. Therefore, we invite you to submit a revised version of the manuscript that addresses the points raised during the review process.

We look forward to receiving your revised manuscript.

Kind regards,

Sonia Brito-Costa, Ph.D.

Academic Editor

PLOS ONE

Journal Requirements:

2. Please note that in order to use the direct billing option the corresponding author must be affiliated with the chosen institute. Please either amend your manuscript to change the affiliation or corresponding author, or email us at plosone@plos.org with a request to remove this option.

Reviewers' comments:

Reviewer's Responses to Questions

**Comments to the Author**

1. Is the manuscript technically sound, and do the data support the conclusions?

Reviewer #1: Partly

2. Has the statistical analysis been performed appropriately and rigorously? 

Reviewer #1: N/A

3. Have the authors made all data underlying the findings in their manuscript fully available?

Reviewer #1: Yes

4. Is the manuscript presented in an intelligible fashion and written in standard English?

Reviewer #1: No

5. Review Comments to the Author

Reviewer #1: I think this paper is very significant and contemporary and the paper includes important aspects. But there are many shortfalls and I do believe this paper needs major correction and almost all of the sub-headings need to rewrite and some sub-heading should be added.

Length: The total length of the article is roughly 70 pages. Although there is no explicit requirement concerning the length of an article, in general, there is an implicit assumption that a length shorter than 6,000 words shall render it difficult to offer a substantial argumentation of the claims raised in an article. The optimal length of an English academic paper is roughly 8,000 words. Indeed, this is simply a general, impressionistic statement which approximates a general guideline to authors. It by no means requires that the author should follow it strictly. Length, by the same token, is not equal to quality. Nevertheless, in the case of this article, a considerably shorter length has apparently done less harm than good, where the author can develop his points in full to the due level of sophistication and clarity – an issue to be backed up by evidences articulated thereafter in this decision report.

6. PLOS authors have the option to publish the peer review history of their article (what does this mean?). If published, this will include your full peer review and any attached files.

Reviewer #1: **Yes: **Taha Husain

---

## [Author Response · Author response to Decision Letter 0]

28 Apr 2022

Dear Dr. Brito-Costa,

Thank you for giving us the opportunity to submit a revised version of our manuscript “The Berlin Misophonia Questionnaire Revised (BMQ-R): Development and validation of a symptom-oriented diagnostical instrument for the measurement of misophonia”. We would also like to thank you and the reviewer for the time and effort you dedicated on giving us feedback. In our revised manuscript, we tried to incorporate every aspect made by the reviewer. Below, you find a response to each of the reviewer’s comments. 

1. Is the manuscript technically sound, and do the data support the conclusions?

Reviewer 1: Partly. 

Author response: We double-checked our methodological approach, our statistical analyses and the conclusions drawn. We believe that our data analyses support the conclusions made. Could you give us an example on where the manuscript is not technically sound and which conclusions are only partly supported by the analyses? 

2. Has the statistical analysis been performed appropriately and rigorously? 

Reviewer 1: N/A. 

Author response: In the course of double-checking, we also re-calculated all of our statistical analyses and corrected minor errors in the manuscript, which do not affect interpretation or conclusions made before. These numerical errors were mainly based on rounding issues and errors in the specification of the maximum likelihood estimators CFA models. We are confident that our analyses are appropriate and correct. 

4. Is the manuscript presented in an intelligible fashion and written in standard English?

Reviewer 1: No. 

Author response: We revised the entire manuscript linguistically. To our knowledge, there are no more typographical or grammatical errors. 

5. Review Comments to the Author

Reviewer 1: I think this paper is very significant and contemporary and the paper includes important aspects. 

Author response: Thank you!

Reviewer 1: […] there are many shortfalls and I do believe this paper needs major correction and almost all of the sub-headings need to rewrite and some sub-heading should be added.

Author response: We would really like to address and correct the raised shortfalls. However, we need to know the specific aspects you have in mind and would therefore like to ask you to specify the detected shortfalls. 

Regarding the sub-headings, we reformulated and reorganised mainly those in the theoretical background of the study. While we agree that the headings in the theoretical part indeed needed to be revised, we respectfully disagree regarding the other headings. The idea behind the subheadings labelled as diagnostic criteria/areas is to give a better overview of the complex structure of the questionnaire and to consequently refer to them in the results, too. We argue that this provides an essential orientation and, in our opinion, other headings would not suit to follow the paper, but impair the understanding tremendously. 

Reviewer 1: The total length of the article is roughly 70 pages. […] in the case of this article, a considerably shorter length has apparently done less harm than good, where the author can develop his points in full to the due level of sophistication and clarity – an issue to be backed up by evidences articulated thereafter in this decision report

Author response: We agree that the paper is very long and therefore made some major changes to address this issue. In total, we were able to remove five pages and about 1400 words. The final manuscript length is about 12000 words (excluding the abstract, tables, figures, and references). In our opinion, a much shorter article would not meet the requirements for reporting materials and methods and would not allow a full understanding of the test development and comprehensive validation study. A comparable article about a scale development for measuring misophonia, which was recently published in Frontiers, with fewer dimensions and a considerably less extensive validation study is about the same length as our revised article (cf. Rosenthal et al., 2021; Development and Initial Validation of the Duke Misophonia Questionnaire). If you see the potential to further shorten the manuscript, we would appreciate more information on what parts are concerned. Below, you find a description of what we have shortened in the revision.

Firstly, we shortened the introduction so that it concisely describes the current literature and the research gap and purpose of our work. Since our submission in 2021, new instruments have been developed and the article must address the alterations in key literature. That is why we needed to include small new paragraphs, however, we managed to rewrite the remainder, so that the paragraph “Existing instruments for measuring misophonia” is shorter than before. Secondly, we described our new measurement approach and the corresponding hypotheses more concisely. Most of the shortening was done in the methods and results sections. We described the many instruments that we used as short as possible and removed additional information. The CTCM(-1) modelling approach for aversive reactions has been removed. In the discussion, we shortened the paragraph describing model misspecifications and the implications from the CTCM(-1) model. Lastly, we removed the conclusions regarding RDoC and HiTOP.

---

## [Editor Report · Decision Letter 1]

23 May 2022

The Berlin Misophonia Questionnaire Revised (BMQ-R): Development and validation of a symptom-oriented diagnostical instrument for the measurement of misophonia

PONE-D-21-25396R1

Dear Dr. Remmert,

We’re pleased to inform you that your manuscript has been judged scientifically suitable for publication and will be formally accepted for publication once it meets all outstanding technical requirements.

Kind regards,

Sónia Brito-Costa, Ph.D.

Academic Editor

PLOS ONE
---

## [Editor Report · Acceptance letter]

10 Jun 2022

PONE-D-21-25396R1 

The Berlin Misophonia Questionnaire Revised (BMQ-R): Development and validation of a symptom-oriented diagnostical instrument for the measurement of misophonia 

Dear Dr. Remmert:

I'm pleased to inform you that your manuscript has been deemed suitable for publication in PLOS ONE. Congratulations! Your manuscript is now with our production department. 

Kind regards, 

on behalf of

Dr. Sónia Brito-Costa 

Academic Editor

PLOS ONE